



# Brown carbon absorption in the red and near infrared spectral region

András Hoffer[1], Ádám Tóth[2], Mihály Pósfai[2], Chul Eddy Chung[3], András Gelencsér[1,2]

[1]MTA-PE Air Chemistry Research Group, Veszprém, P.O. Box 158, H-8201, Hungary
[2]Department of Earth and Environmental Sciences, University of Pannonia, Veszprém, P.O. Box 158, H-8201, Hungary
[3]Division of Atmospheric Sciences, Desert Research Institute, Reno, NV 89512, USA

*Correspondence to*: A. Gelencsér (gelencs@almos.uni-pannon.hu)

**Abstract.** Black carbon aerosols (BC) have been conventionally assumed to be the only light-absorbing carbonaceous particles in the red and near-infrared spectral regions of solar radiation in the atmosphere. Here we report that contrary to the conventional belief tar balls (a specific type of organic aerosol particles from biomass burning) do absorb red and near infrared radiation significantly. Tar balls were produced in a laboratory experiment and their chemical and optical properties were measured. The absorption of these particles in the range between 470 and 950 nm was measured with an aethalometer which is widely used to measure atmospheric aerosol absorption. We find that the absorption coefficient of tar balls at 880 nm is more than 10% of that at 470 nm. The considerable absorption of red and infrared light by tar balls also follows from their relatively low absorption Ångström coefficient (and significant mass absorption coefficient) in the spectral range between 470 and 950 nm. Our results support previous finding that tar balls may play an important role in global warming. Due to the non-negligible absorption of tar balls in the infrared region the absorption measured in the field at higher wavelengths may not solely due to soot particles.

## 1 Introduction

In atmospheric science, black carbon (BC) aka soot aerosols have been conventionally assumed to be the only light-absorbing carbonaceous particles in the red and near-infrared spectral regions of solar radiation in the atmosphere. Organic aerosols (OAs) are currently treated as either being weak absorbers of sunlight in the UV/blue region or having no solar absorption in radiation models (Myhre et al., 2013). Light-absorbing organic aerosols are also known as brown carbon (BrC) since they absorb blue light significantly but have practically zero absorption in the red band, yielding brownish colors (Andreae and Gelencsér, 2006). The distinction between the light absorption by BC and BrC in field and laboratory studies has relied on the explicit assumption that no other carbonaceous particle type except BC absorbs solar radiation at the wavelength of ~700 nm or larger (Bahadur et al., 2012; Kirchstetter and Thatcher, 2012; Saleh et al., 2014; Lu et al., 2015). This common assumption has been used in spite of the



finding by Alexander et al. (2008) who showed a sizable absorption by a specific class of BrC at longer wavelengths. Alexander et al. (2008) named these BrC particles "Brown Carbon Spheres" indicating that the morphology of these particles is similar to that of tar balls. Alexander et al. (2008) derived the absorption using high spatial resolution electron energy-loss spectroscopy (EELS) which is not a direct method for absorption

measurements.

The sources of atmospheric BrC are manifold, ranging from biomass burning emissions to secondary formation in photochemical reactions yielding absorbing particles of various absorption efficiencies (Limbeck et al., 2003; Lukács et al., 2007). Tar balls are widespread in biomass burning smoke (Pósfai et al., 2004; Adachi and Buseck, 2011). These particles clearly belong to the class of BrC and not to BC, as they are distinctly different from BC in

their morphology and other definition properties (Petzold et al., 2013). Tar balls are typically present as spherical solid particles with diameters in the range of 25–500 nm, and can be readily identified by transmission electron microscopy (TEM-EDS), as against BC particles which always have fractal-like morphology. Both tar ball and BC particles are refractory as they can withstand the high vacuum and the irradiation by the electron beam in the TEM indefinitely. As far as elemental composition is concerned, fresh tar balls are nearly homogeneous mixtures of

carbon and oxygen at a molar ratio of about 10:1 as determined by TEM-EDS (Pósfai et al., 2004). While BC particles are primarily formed in fossil fuel combustion and in the flaming stages of biomass fires, tar balls are abundant in relatively aged smoke plumes from smoldering biomass fires (Pósfai et al., 2004).

Recently Tóth et al. (2014) have demonstrated that tar balls very similar to those observed in the atmosphere can be directly produced in the laboratory from liquid tar obtained by the dry distillation of wood chops. Based on these

laboratory experiments the authors postulated that during biomass combustion tar ball particles are generated by the direct ejection of liquid tar droplets from the pores followed by a thermal shock in the fire zone and further atmospheric aging in biomass smoke plumes. This method allows us to directly measure the optical properties of tar balls without the interference of a multitude of other combustion particles. Hoffer et al. (2016) measured the optical properties of tar balls generated in the laboratory only up to 652 nm wavelength, thus the absorption characteristics

of tar balls in the IR region were not discussed in that paper. In this study we used the very same method to generate tar ball particles in the laboratory as before, but the absorption characteristics of the tar balls were also measured directly with a 7-wavelength aethalometer at the wavelengths of 880 nm and 950 nm. This is the first direct experimental measurement of the IR absorption of 'pure' BrC-type particles (tar balls) that are abundant in biomass burning plumes but are definitely not BC.

**2 Experimental procedure**

In this study tar balls were generated in an experimental setup similar to that used by Hoffer et al. (2016). Briefly, liquid tar was produced from dry distillation of 2 different wood types (*Robinia pseudoacacia* (black locust) and *Picea abies* (Norway spruce)) similar to that described in Tóth et al. (2014). The obtained liquid distillate consisted of an aqueous phase and an oily phase. During the experiments, only the aqueous phase was used. This phase was

concentrated and taken up with methanol. Droplets were then generated from the methanol solution by an ultrasonic





atomizer (1.6 MHz, Exo Terra Fogger, PT2080, Rolf C. Hagen Corp), and they were aged by heat at 650°C for about 1 second, using a tube furnace (Carbolite, MTF 10/25/130). This ageing process is of utmost important as it influences the chemical composition and optical properties of the formed particles (Hoffer et al., 2016) During the particle generation the system was continually rinsed with $N_2$ containing 4% (v/v) $O_2$. The particles were then dried

and diluted with dry filtered air. Before the optical measurements a PM1 cyclone (SCC 2.229) was deployed to remove the large particles (Dp> ~500 nm).

The absorption properties of the particles were measured with two optical instruments at different wavelengths. The absorption coefficient at 467 nm, 528 nm and 652 nm was measured with a continuous light absorption photometer (CLAP) with a time resolution of 5 seconds. In order to extend the measurements of the optical properties into the

longer wavelength range (at 880 and 950 nm) an aethalometer (MAGEE AE42-7) was applied with a time resolution of 2 minutes. The CLAP data whose measurement principle is similar to that of PSAP are corrected according to Bond et al. (1999) and Ogren (2010) with the data processing algorithm applied by NOAA. The aethalometer data were corrected according to the Weingartner correction scheme (Weingartner et al., 2003) and also by the Schmid correction (Schmid et al., 2006). The latter largely affects the Ångström exponent, whereas the former correction

scheme has no effect on the Ångström exponent (Coen et al., 2010). In the correction we used the absorption coefficient measured by the CLAP at 528 nm as the reference value, keeping in mind that its reliability in the absolute scale is about 25% (Schmid et al., 2006).

The scattering coefficient was measured by a TSI 3563 nephelometer at 3 different wavelengths (450, 550 and 700 nm) with a time resolution of 5 seconds. The collected raw data were corrected according to Anderson and Ogren

20 (1998).

During the experiment the size distribution between 7 and 800 nm was measured with a DMPS designed by the University of Helsinki.

The morphology and the elemental composition of the tar balls collected on TEM grids (lacey Formvar/carbon TEM copper grid of 200 mesh, Ted Pella Inc., USA) were studied in brightfield TEM images obtained using a Philips

CM20 TEM operated at 200 kV accelerating voltage. An ultra-thin-window Bruker Quantax X-ray detector was attached to the electron microscope that allowed the energy-dispersive X-ray analysis (EDS) of the elemental compositions of individual particles.

## 3 Results

### 3.1 Morphology and size distribution of generated tar balls

Figure 1 shows that the morphology of the generated particles is very similar to that of the freshly formed tar balls (Adachi and Buseck, 2011), as the particles are perfect or slightly distorted spheres. The size distribution measured with a DMPS was similar to that obtained previously by Hoffer et al. (2016). The volume size distribution consists of a double peak; the larger is at 116 nm and 139 nm in the case of black locust and Norway spruce, respectively (see Figure 2). In this mode more than 96% of the particulate mass can be found. There is no difference in the molar





C/O ratio between tar balls generated from the two wood types; it varies between 8.1 and 11.6 with an average of 9.3 as determined from TEM-EDS analyses.

## 3.2 Absorption properties of the generated tar balls

The absorption Ångström exponents (AAE) of the tar balls generated from the aqueous phase of the dry distillate from black locust and Norway spruce measured by the CLAP between 467 and 652 nm are 2.9 and 3.2, respectively (see Figure 3). Similar values are obtained from the aethalometer data between 470 and 590 nm using the Weingartner correction scheme (Weingartner et al., 2003). The AAE in the same wavelength range (470–590 nm) for the black locust and Norway spruce calculated from the aethalometer data and corrected according the Schmid correction scheme (Schmid et al., 2006) are 3.2 and 3.5, respectively.

The absorption measured at 370 nm by the aethalometer was, however, significantly lower than it would be predicted from AAE curve fitting. That is why the absorption data at this wavelength was omitted in the AAE calculations. Figure 3 shows that the AAE between 470 and 950 obtained by curve fitting is 3.1–3.2 and 3.4–3.6, depending on the applied correction scheme (Schmid correction gives higher values) for the black locust and Norway spruce, respectively. The error bars on Figure 3 indicate the estimated 25% uncertainty of the absorption measurements for all wavelengths. This value was reported for a one-wavelength PSAP by Schmid et al. (2006). Here we note that Chow et al. (2009) reported higher and wavelength-dependent uncertainties between the filter-based and photoacoustic methods (17–69%, larger differences at higher wavelengths), but the difference in AAE measured with different instruments were below 25%.

Since tar balls in the ambient atmosphere might have a size distribution different from those generated in our lab, the AAE of the atmospheric tar balls might also be somewhat different since AAE depends on the size distribution as well. The particle generation procedure used in the present study, as well as the measurement of the optical properties were similar to those used by Hoffer et al. (2016). These authors calculated the AAE for tar balls using the ambient size distribution of these particles determined by Pósfai et al. (2004) and found that the AAE decreased from ~2.9 to ~2.4 in the range between 462 and 652 nm. Since the size distribution of the generated particles in the present study was similar to that obtained by Hoffer et al. (2016) and the AAE of the tar balls in the same wavelength range (462–652 nm) were also similar, the AAE of freshly formed tar ball particles between 470 and 950 nm might be somewhat lower (by about 20% based on the results by Hoffer et al., 2016) under ambient conditions than suggested by our calculations. On the other hand, Chow et al. (2009) showed that the AAE of ambient aerosol obtained from a photoacoustic analyzer (PA) is noticeably higher (by 14–23%) than that obtained using filter-based instruments such as the aethalometer. In view of this, we propose that ambient freshly formed tar balls likely have an AAE of 2.7 ~ 3.6 in the wavelength range from 470 to 950 nm. (The value 2.7 is the lowest AAE value of the generated tar balls from oak reported by Hoffer et al., 2016, whereas the upper value of the given range is the highest AAE value obtained in the present study.)

As Figure 3 demonstrates the absorption of tar balls is non-negligible in the near IR range. The absorption coefficient at 880 nm is more than 10% of that measured at 470 nm for both wood types, undermining the common assumption that all BrC particles have zero absorption at 880 nm. Even at 950 nm, the absorption coefficient is





about 10% of that at 470 nm. The mass absorption efficiency of tar balls in the near IR range is expected to be substantial as well, given that the mass absorption efficiency of tar balls was estimated to be in the range of 0.8–3 $m^2g^{-1}$ at 550 nm by Hoffer et al. (2016). As Chow et al. (2009) demonstrated absorption measured by filter-based instruments poses significant uncertainties and so follow-up studies are desired to reduce the uncertainty of

estimated tar ball absorption.

In spite of the uncertainties in the measurements of the absorption and other parameters, we estimated the index of refraction of the generated tar balls at different wavelengths. In order to address the measurement uncertainties nigrozin particles were generated and measured with the same setup used for tar ball measurements. The measured absorption and scattering coefficients at 652 and 633 nm, respectively, were compared with those calculated using

the size distribution and the index of refraction of nigrozin at 633 nm wavelength (Pinnick et al., 1973). The obtained correction factors were applied for the measured scattering and absorption coefficients of tar balls, which together with the size distribution served as input parameters for the inverse Mie calculations (Guyon et al., 2003). It was assumed that the same correction factors apply for the other wavelengths as well. For the calculations the absorption and scattering coefficient were extrapolated to the given wavelength, if it was necessary. Table 1

summarizes the average refractive index of tar balls at different wavelengths. The obtained index of refraction data between 470 and 652 nm are very close to those obtained previously for turkey oak (Hoffer et al., 2016). The imaginary parts of the refractive index of tar balls produced from different wood types are very similar to each other at higher wavelength too. Here we note that, since tar balls form during burning and/or pyrolytic processes, in which the temperature might highly affect the composition and consequently the optical properties of the formed particles,

the data in the table are not necessarily characteristic for every atmospheric tar ball particle. In our experiments we produced tar balls whose elemental composition and morphology matches with those of atmospheric tar balls reported for the first time by Pósfai et al., (2004).

### 3.3 Estimated contribution of tar balls to the absorption at K-puszta station

The objective of this section is to assess the possible contribution of tar ball particles to the absorption at a given station (where the size distribution and the number share of tar balls were measured previously by TEM analyses (Pósfai et al., 2004)), based on the novel finding that absorption of these particles in the IR range is non-negligible. During winter the K-puszta station is affected by biomass smoke from domestic heating. Pósfai et al. (2004) investigated aerosol samples collected at this station and identified tar ball particles by electron microscopy. The

concentration of levoglucosan is also elevated at the station during winter (Puxbaum et al., 2007).

In order to estimate the contribution of tar balls to the absorption measured at K-puszta station we calculated the absorption coefficient of a tar ball population with a Mie code at 652 nm and compared the calculated values with the measured absorption coefficient at this wavelength. The index of refraction of tar balls at 652 nm ($1.82 - 0.15i$) was taken from Hoffer et al. (2016). For the comparison we used the measured absorption coefficient (CLAP, at 652

nm) as well as the measured particle number concentration (DMPS between 7 and 800 nm) at K-puszta during 10



days between 07.01.2014 and 27.01.2014. The shape of the size distribution of tar balls was taken from Pósfai et al. (2004) as it was determined from tar balls collected at K-puszta. Furthermore we assumed that the number share of tar ball particles is 20%, as Pósfai et al. (2004) reported that the contribution of tar ball particles to the total number concentration varies between 0 and 40% in K-puszta. If we consider that only tar balls and soot are the absorbing

components at 652 nm, the contribution of the tar balls to the absorption is 17–38%, on average 29%. Even if we consider that only 5% of the particles are tar balls, the contribution to the absorption is still 4–9% (on average 7%) at 652 nm. This also indicates that the contribution of the tar balls to the absorption at higher wavelengths might be significant too, since the AAE of tar balls is 2.7–3.6, which means that the absorption do not decrease steeply towards the IR range of the spectrum. (Based on the comparison of the extrapolated measurement data with the

estimated absorption coefficient for tar balls obtained for 880 nm, the contribution of tar balls to the absorption at 880 nm is 5–19% at the K-puszta station.).

Here we note that the AAE measured at the station (1.58–1.88, on average 1.71 between 467 and 652 nm) during the investigated period (10 days, between 07.01.2014 and 27.01.2014) is very close to that (AAE=1.76 between 467 and 652 nm) obtained by the estimation assuming soot (AAE=1) and tar ball (AAE=3.15) as the only absorbing

components, the latter causing on average 29% of the absorption at 652 nm. Since the contribution of humic-like substances to the absorption (few per cent at 550 nm, Hoffer et al., 2006) is incorporated in the measured AAE, the estimated contribution of the tar ball particles might be considered as an upper limit. If we assume that the contribution of tar ball particles to the total particle number concentration is only 5% (which also means that the contribution of tar balls to the absorption at 652 nm is 7%, see above), the calculated AAE, assuming soot and tar

ball as the only absorbing components, decreases significantly, it is on average 1.19 between 467 and 652 nm. Since the measured AAE is higher than the calculated value, the contribution of 7% to the absorption (5% in number concentration) can be considered as a lower value during the investigated period.

## 4 Summary

We have used a CLAP to measure the absorption of tar ball particles between 462 and 652 nm and an aethalometer

between 470 and 950 nm. The aethalometer has two measurement channels in the infrared region, 880 and 950 nm, thus allowing for direct measurement of the light absorption in the red-infrared part of the spectrum. The Absorption Ångström Exponent (AAE) of tar balls over 470 ~ 950 nm is in the range between 2.7 and 3.6, but more importantly, the absorption coefficient at 880 nm exceeds 10% of that at 470 nm for both wood types (Figure 3), clearly disproving the common assumption that all BrC particles have zero absorption at 880 nm.

The determination of the contribution of BrC to aerosol absorption (Bahadur et al., 2012; Kirchstetter and Thatcher, 2012; Saleh et al., 2013; Lu et al., 2015) has been based on the explicit assumption that BrC has zero absorption at the wavelength of 700 nm or larger. The findings of the present study strongly challenge this common assumption. One of the resulting implications may be that the role of BC—a significant fraction of which is derived from fossil fuel combustion (diesel soot)—is likely overestimated in global radiative forcing estimates if the aerosol absorption

in the red and near-infrared spectrum is attributed exclusively to BC. Our results support the finding by Alexander et





al., (2004) as well as Jacobson et al., (2014), that tar balls effectively absorb solar radiation, and because of their widespread atmospheric occurrence they also have an important role in global warming.

**Acknowledgements**

The authors thank NOAA ESRL laboratory and the University of Helsinki for their support in data management and
5 the size distribution measurements. This project has received funding from the European Union's Horizon 2020 research and innovation programme under grant agreement No 654109. This study was funded by the National Science Foundation (AGS-1455759).



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

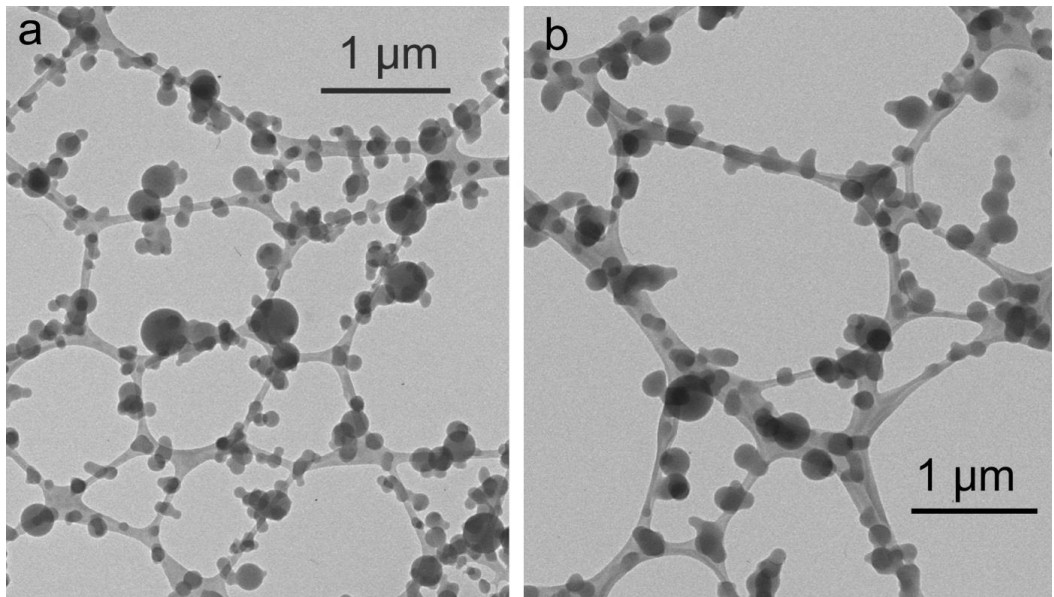

**Figure 1. TEM images of tar balls generated from (a) black locust (*Robinia pseudoacacia*) and (b) Norway spruce (*Picea abies*).**





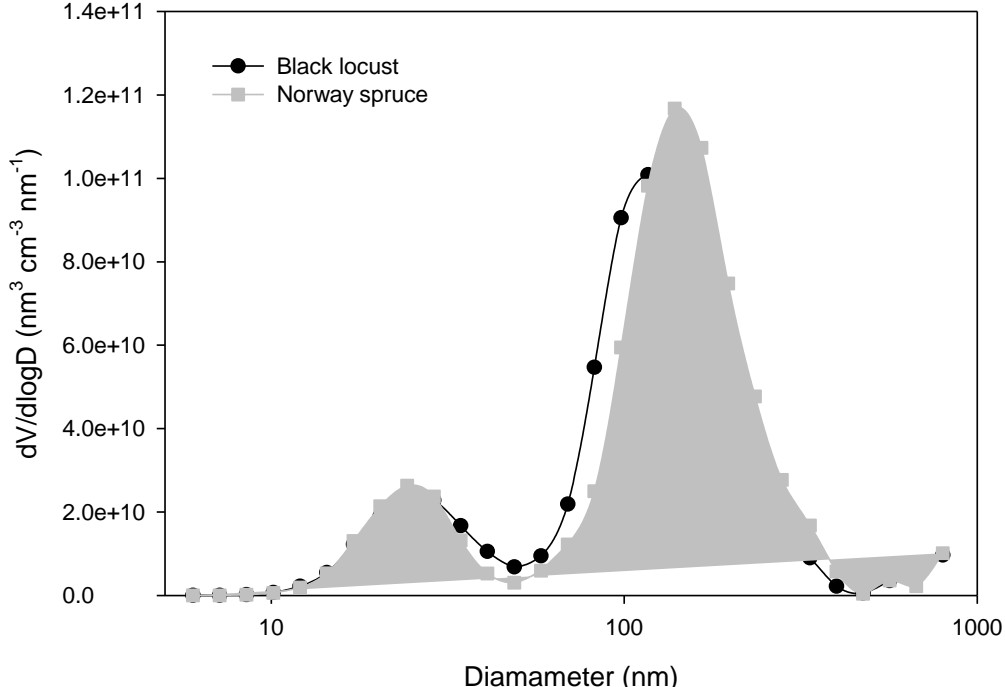

15    **Figure 2. Examples for the measured size distribution of laboratory generated tar balls from the dry distillate of black**
16    **locust and Norway spruce.**

17





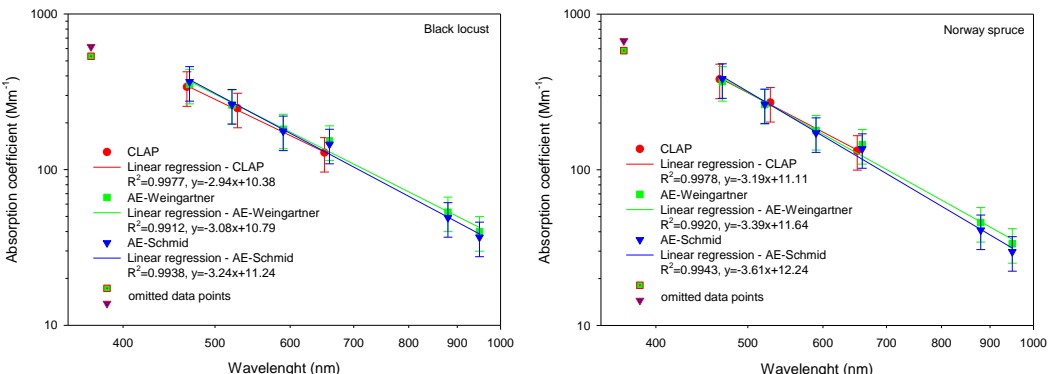

18  **Figure 3. Absorption Ångström exponent of tar balls prepared from the liquid distillate of black locust and Norway**
19  **spruce. The linear equations are calculated for the log A vs. log λ values.**

20


| Absorption instrument | Wavelenght (nm) | Black locust | | Norway spruce | | Oak (Hoffer et al., 2016) | |
|---|---|---|---|---|---|---|---|
| | | Re | Im | Re | Im | Re | Im |
| CLAP | 467 | 1.86 | 0.34 | 1.88 | 0.33 | 1.84 | 0.27 |
| CLAP | 550 | 1.86 | 0.25 | 1.88 | 0.24 | 1.84 | 0.21 |
| CLAP | 652 | 1.77 | 0.18 | 1.82 | 0.16 | 1.82 | 0.15 |
| CLAP | 880 | 1.64 | 0.10 | 1.84 | 0.09 | | |
| CLAP | 950 | 1.61 | 0.09 | 1.85 | 0.08 | | |
| Aethalometer | 880 | 1.64 | 0.09 | 1.83 | 0.08 | | |
| Aethalometer | 950 | 1.60 | 0.07 | 1.83 | 0.07 | | |

21

22  **Table 1. The average Real (Re) and imaginary (Im) part of the complex index of refraction of laboratory generated tar**
23  **balls from different wood types.**

24