# Peer review of "Brown carbon absorption in the red and near infrared spectral region"

_Atmospheric Measurement Techniques, 2016_

## Referee Comment (RC1) · Anonymous Referee #2 · 3 Feb 2017

This paper discusses measured absorption properties of tar balls from ∼652 nm to 950 nm wavelength, following from Hoffer et al.'s (2016) paper on the same topic from 467-652 nm. The paper contributes unique information to the literature, so I think it should be published following minor revisions listed below.

1) Abstract. I don't think that absorption by tar balls in the red and near-infrared part of the solar spectrum is "contrary to conventional belief" as illustrated in Alexander et al. (2008). The issue has been a question of the magnitude of absorption. You can argue that most (but not all) models have assumed no absorption – however, even then, most models have not included tar balls at all. So you could say that most models have not included tar balls and of those that have, most have not accounted for red and infrared absorption. Please modify the abstract to more clearly delineate this.

2) Page 1, Line 26. State "in most all radiation models" rather than "in radiation models" Since some radiation models (e.g., in Jacobson et al., 2014) have included organic absorption in the red and near infrared.

3) Page 5. Why nigrozin?

4) Page 6. Please clarify why if tar balls are only 5% of the particle number, they represent 7% of absorption. To do this, you can use an example with a monodisperse distribution of BC particles at their mean size and one of tar ball particles at their mean size Provide the number of particles, the cross-sectional area of particles, and the single-particle absorption efficiency of particles at each size and calculate the extinction coefficient of each monodisperse size distribution.

5) Summary. Please clarify that Alexander et al. did not examine the impact of tar balls on global warming.

---

## Referee Comment (RC2) · Anonymous Referee #1 · 9 Feb 2017

General comments The experimental study presented by authors expands the spectral range of measured spectral absorption of the brown carbon (BrC) up to 950 nm, while previous studies have been limited by only visible range (up to 652 nm). This result one may seem new and useful, but somewhat particularistic, but in fact it is significant. The existent assumption while estimating atmosphere absorption is neglecting of the BrC particles absorption at wavelengths 700nm and more. Moreover some used models haven't included absorption by BrC particles. On the base of obtained data authors put forward the reasons that the contribution of BrC to global aerosol absorption is likely higher than previously estimated, implying a more significant role of biomass burning in global radiative forcing and regional radiative effects.

The paper is written clearly, each paragraph in this research is justified. The list of references demonstrates the authors' appropriate knowledge of the state of the art in

the problem.

The paper is recommended to be published. I would like to recommend only one minor, but may be important addition.

Specific comments

The most important experimental results of this research is the measured data of BrC tar ball absorption including partially the near IR range. From my point of view it would be necessary to make these data available to the readers. Nevertheless the only Angstrom exponent in the double logarithmic scale is given (Fig.3). It is very scarce, somewhat qualative, presentation, and it is not only because of a used scale. The Angstrom exponent taken over so large range isn't a very accurate characteristic of spectral behavior (See for instance presentation of the AERONET data). May be more informative way is to include the experimental spectral absorption data as tables.

---

## Short Comment (SC1) · 9 Feb 2017

This manuscript presents interesting findings on light absorption by tar balls. The authors present evidence for absorption at long visible and IR wavelengths, however, their claim that this finding is "contrary to the conventional belief" (P.1, L.12) is not accurate. There are at least two studies that have shown this before, Alexander et al. (2008) and Saleh et al. (2014), both cited in the manuscript. The authors cite Saleh et al. (2014) among studies that explicitly assumed that non-BC carbonaceous aerosols do not absorb at wavelengths > 700 nm (P.1, L. 30). This is not true. In fact, Saleh et al. (2014) presented evidence for extremely low volatility organic compounds (ELVOCs) in biomass-burning emissions being highly absorptive in the visible spectrum and near IR. The imaginary parts of the refractive indices reported in Figure 2 of Saleh et al. (2014) are in good agreement with those reported in this study for tar balls. There is a

strong argument to be made that tar balls constitute a fraction of Saleh et al.'s ELVOCs. I believe that a comparison of the findings of this study with those of Saleh et al. and Alexander et al. should be discussed in this paper.

---

## Referee Comment (RC3) · Anonymous Referee #3 · 15 Feb 2017

This paper presents the wavelength dependence of the absorption Angstrom exponent and complex refractive index of laboratory generated tar balls from two wood types. The retrieved values cover the 467-959 nm wavelength range extending previous published values to the nIR region. The results are novel and of interest for atmospheric modeling and therefore within the scope of AMT. I recommend the publication in AMT after the following issues has been addressed:

Specific comments

1. According to the manuscript, the Continuous Light Absorption Photometer (@ 467 nm, 528 nm, and 652) and aethalometer (@880 and 950 nm) provide the wavelength dependence of the absorption coefficient. Scattering coefficients are measured by a TSI 3563 nephelometer at 450, 550, and 700 nm but they are not provided in the paper.

Please, include those data. Does it mean that the scattering coefficients of the tar balls are not measured at 880 and 950 nm but only the absorption coefficients? Please, specify this point.

2. Page 5, lines 5-10:

' In order to address the measurement uncertainties nigrozin particles were generated and measured with the same setup used for tar ball measurements. The measured absorption and scattering coefficients at 652 and 633 nm, respectively, were compared with those calculated using the size distribution and the index of refraction of nigrozin at 633 nm wavelength (Pinnick et al., 1973). "

How is the scattering coefficient (633 nm) of nigrozin particles measured? According to page 3, lines 18-20, the TSI 3563 nephelometer provides scattering coefficients at 450, 550, and 700 nm. Is it measured by another instrument?

3. Page 5, lines 10-13:

"The obtained correction factors were applied for the measured scattering and absorption coefficients of tar balls, which together with the size distribution served as input parameters for the inverse Mie calculations (Guyon et al., 2003). "

It is mentioned that the uncertainties in the measured absorption coefficients are around 25%. Are the obtained corrections factors within the estimated 25% error? Please, provide the correction factors. Those values are needed to have an indication of the accuracy of the measured values.

4. Page 5, lines 14-15.

"It was assumed that the same correction factors apply for the other wavelengths as well. For the calculations the absorption and scattering coefficient were extrapolated to the given wavelength, if it was necessary."

It should be clearly stated which refractive indices are obtained from the measured

scattering and absorption coefficients and which from extrapolated values. In the extrapolation procedure you are assuming a linear dependence on the scattering and absorption coefficients with the wavelength. Is that correct? Please, specify.
* * *

---

## Author Comment (AC1) · 4 Apr 2017

Response to Interactive comment of Referees

Anonymous Referee #2

1) Abstract. I don't think that absorption by tar balls in the red and near-infrared part of the solar spectrum is "contrary to conventional belief" as illustrated in Alexander et al. (2008). The issue has been a question of the magnitude of absorption. You can argue that most (but not all) models have assumed no absorption – however, even then, most models have not included tar balls at all. So you could say that most models have not included tar balls and of those that have, most have not accounted for red and infrared absorption. Please modify the abstract to more clearly delineate this.

[Figure]

P.1, L.10-13. The abstract has been modified:

Black carbon aerosols (BC) have often been assumed to be the only light-absorbing carbonaceous particles in the red and near-infrared spectral regions of solar radiation in the atmosphere. Here we report that tar balls (a specific type of organic aerosol particles from biomass burning) do absorb red and near infrared radiation significantly.

2) Page 1, Line 26. State "in most all radiation models" rather than "in radiation models" Since some radiation models (e.g., in Jacobson et al., 2014) have included organic absorption in the red and near infrared.

P.1, L.27. The sentence has been modified. solar absorption in most radiation models (Myhre et al., 2013)

3) Page 5. Why nigrozin?

Nigrozin is a standard, commercially available, stable substance with a well-known index of refraction at a single wavelength (633nm, Pinick et al., 1973). It was used for the verification of our measurement system and for the assessment of the potential bias of our measurements. To do this, we generated nigrosin droplets from solution and measured them by the very same instrumental setup that was used for the tar ball measurements. It is also important that the single scattering albedo (SSA) of nigrozin droplets is close to that of the tar ball particles studied.

4) Page 6. Please clarify why if tar balls are only 5% of the particle number, they represent 7% of absorption. To do this, you can use an example with a monodisperse distribution of BC particles at their mean size and one of tar ball particles at their mean size. Provide the number of particles, the cross-sectional area of particles, and the single-particle absorption efficiency of particles at each size and calculate the extinction coefficient of each monodisperse size distribution.

Similar calculation was presented in the manuscript in the case of the spherical tar balls: We used the index of refraction of tar balls at 652 nm and calculated (by a Mie

code) the single particle absorption efficiency for the size bins of the size distribution measured for ambient tar balls by Pósfai et al., (2004). During the calculations the number concentration of tar balls were adjusted between 20% and 5% of the total number concentration measured by a DMPS at the K-puszta station, but the size distribution of tar balls was kept constant. The derived absorption coefficient of tar balls were then compared to the measured absorption coefficient at the site. The absorption of BC was inferred by difference.

Following the recommendations of the reviewer if a Mie code is used to calculate the absorption of soot (it means we assume spherical soot particles) with a mass median diameter of 140 nm (Bond et al., 2013) and with the index of refraction of 1.95–0.79i at 652 nm (Bond and Bergstrom 2006), the single particle absorption efficiency at this wavelength is 0.9599.

By taking an index of refraction of tar balls of 1.82–0.15i at 652 nm and a mass median diameter of 220 nm based on Pósfai et al., 2004, we obtain a single particle absorption efficiency of 0.4606 for tar balls. By assuming number concentrations of soot particles and tar ball to be 95 cm–3 and 5 cm–3, respectively, we obtain an absorption coefficient of $1.40\times10$–6 m–1 for soot particles, and of $8.75\times10$–8 m–1 for tar balls. It means that the relative contribution of tar balls to the overall absorption is ∼6 % which is comparable to the values derived from field measurements.

Bond, T. C., and Bergstrom, R. W.: Light absorption by carbonaceous particles: An investigative review, Aerosol Science and Technology, 40, 27-67, 10.1080/02786820500421521, 2006.

Bond, T. C., Doherty, S. J., Fahey, D. W., Forster, P. M., Berntsen, T., DeAngelo, B. J., Flanner, M. G., Ghan, S., Karcher, B., Koch, D., Kinne, S., Kondo, Y., Quinn, P. K., Sarofim, M. C., Schultz, M. G., Schulz, M., Venkataraman, C., Zhang, H., Zhang, S., Bellouin, N., Guttikunda, S. K., Hopke, P. K., Jacobson, M. Z., Kaiser, J. W., Klimont, Z., Lohmann, U., Schwarz, J. P., Shindell, D., Storelvmo, T., Warren, S. G., and Zender, C.

S.: Bounding the role of black carbon in the climate system: A scientific assessment, Journal of Geophysical Research-Atmospheres, 118, 5380-5552, 10.1002/jgrd.50171, 2013.

Pósfai, M., Gelencsér, A., Simonics, R., Arató, K., Li, J., Hobbs, P. V., and Buseck, P. R.: Atmospheric tar balls: Particles from biomass and biofuel burning, Journal of Geophysical Research-Atmospheres, 109, 10.1029/2003jd004169, 2004

5) Summary. Please clarify that Alexander et al. did not examine the impact of tar balls on global warming.

P.7, L.9-11. The sentence has been changed.

Our results support the finding by Alexander et al. (2008) that spherical brown carbon particles effectively absorb near-infrared radiation as well as that by Jacobson et al. (2014) that they also have an important role in global warming.

Anonymous Referee #1

The most important experimental results of this research is the measured data of BrC tar ball absorption including partially the near IR range. From my point of view it would be necessary to make these data available to the readers. Nevertheless the only Angstrom exponent in the double logarithmic scale is given (Fig.3). It is very scarce, somewhat qualative, presentation, and it is not only because of a used scale. The Angstrom exponent taken over so large range isn't a very accurate characteristic of spectral behavior (See for instance presentation of the AERONET data). May be more informative way is to include the experimental spectral absorption data as tables.

The data of Figure 3 are presented in the table of the supplementary material.

R. Saleh rawad@uga.edu

This manuscript presents interesting findings on light absorption by tar balls. The authors present evidence for absorption at long visible and IR wavelengths, however, their

claim that this finding is "contrary to the conventional belief" (P.1, L.12) is not accurate. There are at least two studies that have shown this before, Alexander et al. (2008) and Saleh et al. (2014), both cited in the manuscript. The authors cite Saleh et al. (2014) among studies that explicitly assumed that non-BC carbonaceous aerosols do not absorb at wavelengths > 700 nm (P.1, L. 30). This is not true. In fact, Saleh et al. (2014) presented evidence for extremely low volatility organic compounds (ELVOCs) in biomass-burning emissions being highly absorptive in the visible spectrum and near IR. The imaginary parts of the refractive indices reported in Figure 2 of Saleh et al. (2014) are in good agreement with those reported in this study for tar balls. There is a strong argument to be made that tar balls constitute a fraction of Saleh et al.'s ELVOCs. I believe that a comparison of the findings of this study with those of Saleh et al. and Alexander et al. should be discussed in this paper.

P.2, L1. The Saleh et al., (2014) paper has been removed from the list of references which explicitly assumed that non-BC carbonaceous aerosols do not absorb at wavelengths > 700 nm. The following sentences have been added to the manuscript:

P.2, L.5-8: Not long ago Saleh et al. (2014) identified extremely low volatile compounds (ELVOC) in biomass burning aerosols and calculated the index of refraction of these compounds. They found that these compounds absorb light in the near infrared range as well, and as BrC components they have an important role in direct radiative forcing.

P.5, L.25-30: The obtained index of refraction in the lower wavelength range (between ~460 and ~650 nm) is near the higher bound of the range found by Saleh et al. (2014) for ELVOC and agrees well at ~950 nm. Since tar balls are produced and/or form in high temperature processes (and also withstand the electron beam in the TEM) they are thermally stable compounds and as those that can be classified as extremely low volatile compounds. On the other hand the obtained index of refraction is somewhat lower (especially at longer wavelengths) than those reported by Alexander et al. (2008).

Anonymous Referee #3

1, According to the manuscript, the Continuous Light Absorption Photometer (@ 467 nm, 528 nm, and 652) and aethalometer (@880 and 950 nm) provide the wavelength dependence of the absorption coefficient. Scattering coefficients are measured by a TSI 3563 nephelometer at 450, 550, and 700 nm but they are not provided in the paper. Please, include those data. Does it mean that the scattering coefficients of the tar balls are not measured at 880 and 950 nm but only the absorption coefficients? Please, specify this point.

Indeed, the scattering coefficients at 880 and 950 nm were not measured but extrapolated from the measured values. To extrapolate the scattering data the scattering Ångström exponent was calculated between 450 and 700 nm and using a power function the scattering coefficients were determined at higher wavelengths. The absorption coefficient was measured by an aethalometer and by a CLAP. To calculate the index of refraction of tar balls at higher wavelengths the extrapolated CLAP data were also used (as indicated in Table 1), since the time resolution of the CLAP instrument was much higher than that of the aethalometer (5 seconds and 2 minutes, respectively). As requested by the reviewer the scattering and the absorption coefficients used as input parameters for the inverse Mie calculations are now given in the Supplement.

2. Page 5, lines 5-10: ' In order to address the measurement uncertainties nigrozin particles were generated and measured with the same setup used for tar ball measurements. The measured absorption and scattering coefficients at 652 and 633 nm, respectively, were compared with those calculated using the size distribution and the index of refraction of nigrozin at 633 nm wavelength (Pinnick et al., 1973). " How is the scattering coefficient (633 nm) of nigrozin particles measured? According to page 3, lines 18-20, the TSI 3563 nephelometer provides scattering coefficients at 450, 550, and 700 nm. Is it measured by another instrument?

Indeed, the scattering was not measured at 633 nm, it was interpolated using the same equation that was used to extrapolate the scattering coefficient at higher wavelengths.

[Figure]

P.5, L.12. The sentence has been modified: The measured absorption and interpolated scattering coefficients at 652 and 633 nm, respectively, were compared with those calculated.

3. Page 5, lines 10-13: "The obtained correction factors were applied for the measured scattering and absorption coefficients of tar balls, which together with the size distribution served as input parameters for the inverse Mie calculations (Guyon et al., 2003). " It is mentioned that the uncertainties in the measured absorption coefficients are around 25%. Are the obtained corrections factors within the estimated 25% error? Please, provide the correction factors. Those values are needed to have an indication of the accuracy of the measured values.

For the absorption coefficient the correction factor obtained from the nigrozin measurements was 1.098, and that for the scattering coefficient was 0.614. The correction factor for the absorption is well within the estimated error of 25%. As regards the scattering correction factor it is important to note that Massoli et al. (2009) found the uncertainty of the scattering coefficient measured by a TSI nephelometer to be $25-30\%$ for particles with single scattering albedo of 0.4, and $16-18\%$ for SSA of 0.5. The single scattering albedo of nigrosin particles were on average 0.41 as calculated from the measured absorption coefficient at 652 nm and from the interpolated scattering coefficient at 633 nm. Since the SSA of tar ball particles at 652 nm (measured absorption and interpolated scattering coefficient) was on average 0.53, the correction may be applied for the tar balls within the 25% uncertainty limit. It should be worthy of note that the correction factors obtained from the nigrozin measurements also address the sizing bias (The volume size distributions of the nigrozin particles consist of a single peak at particle diameter of 166 nm, being quite similar to those obtained for tar balls (116 and 139 nm, depending on the wood type.)

The first paragraph has been added to the supplementary file.

4. Page 5, lines 14-15. "It was assumed that the same correction factors apply for

the other wavelengths as well. For the calculations the absorption and scattering co-efficient were extrapolated to the given wavelength, if it was necessary." It should be clearly stated which refractive indices are obtained from the measured scattering and absorption coefficients and which from extrapolated values. In the extrapolation procedure you are assuming a linear dependence on the scattering and absorption coefficients with the wavelength. Is that correct? Please, specify.

To extrapolate both the absorption and scattering data power functions were applied as indicated above. Table S2 shows the optical input parameters for the Mie calculations indicating which data were measured and which data were obtained by inter- or extrapolation.

Please also note the supplement to this comment:
http://www.atmos-meas-tech-discuss.net/amt-2016-392/amt-2016-392-AC1-supplement.pdf

**Supplement:**

**Supplementary material**

**Brown carbon absorption in the red and near infrared spectral region**

András Hoffer[1], Ádám Tóth[2], Mihály Pósfai[2], Chul Eddy Chung[3], András Gelencsér[1,2]

[1]MTA-PE Air Chemistry Research Group, Veszprém, P.O. Box 158, H-8201, Hungary
[2]Department of Earth and Environmental Sciences, University of Pannonia, Veszprém, P.O. Box 158, H-8201, Hungary
[3]Division of Atmospheric Sciences, Desert Research Institute, Reno, NV 89512, USA

The data of Figure 3 are summarized in Table S1. The CLAP and aethalometer data at similar wavelengths were measured simultaneously.

| | Wavelenght (nm) | Absorption coefficient (Mm$^{-1}$) | | | |
| --- | --- | --- | --- | --- | --- |
| | | Black locust | | Norway Spruce | |
| | | Weingartner | Schmid | Weingartner | Schmid |
| Aethalometer | 370 | 535.57 | 615.98 | 582.45 | 674.15 |
| | 470 | 353.10 | 367.47 | 368.54 | 384.19 |
| | 521 | 260.28 | 262.02 | 263.04 | 264.20 |
| | 590 | 181.75 | 176.73 | 178.64 | 172.64 |
| | 660 | 153.04 | 145.55 | 145.27 | 136.39 |
| | 880 | 53.23 | 49.10 | 45.77 | 40.95 |
| | 950 | 39.93 | 36.76 | 33.49 | 29.76 |
| CLAP | 467 | 340.14 | | 382.03 | |
| | 528 | 247.94 | | 270.54 | |
| | 652 | 128.61 | | 132.66 | |

**Table S1. Absorption coefficient measured by the aethalometer and the CLAP**

For the absorption coefficient the correction factor obtained from the nigrozin measurements was 1.098, and that for the scattering coefficient was 0.614. The correction factor for the absorption is well within the estimated error of 25%. As regards the scattering correction factor it is important to note that Massoli et al. (2009) found the uncertainty of the scattering coefficient measured by a TSI nephelometer to be 25−30% for particles with single scattering albedo of 0.4, and 16−18% for SSA of 0.5. The single scattering albedo of nigrosin particles were on average 0.41 as calculated from the measured absorption coefficient at 652 nm and from the interpolated scattering coefficient at 633 nm. Since the SSA of tar ball particles at 652 nm (measured absorption and interpolated scattering coefficient) was on average 0.53, the correction may be applied for the tar balls within the 25% uncertainty limit.

|        | S467* | A467  | S550  | A550* | S652* | A652  | S700  | A700* | S880* | A880* | S950* | A950* | AE880 | AE950 |
|--------|-------|-------|-------|-------|-------|-------|-------|-------|-------|-------|-------|-------|-------|-------|
| Bl. d1 | 195.2 | 363.2 | 133.7 | 235.7 | 78.1  | 140.8 | 64.3  | 115.1 | 34.3  | 60.1  | 27.8  | 48.4  | 52.6  | 39.3  |
| Bl. d2 | 200.9 | 385.6 | 138.7 | 249.1 | 80.6  | 148.3 | 66.4  | 121.0 | 35.5  | 62.8  | 28.8  | 50.4  | 61.4  | 45.9  |
| Bl. d3 | 206.2 | 377.5 | 142.8 | 244.7 | 83.8  | 146.2 | 69.2  | 119.4 | 37.4  | 62.3  | 30.4  | 50.1  | 68.1  | 51.3  |
| Bl. d4 | 204.9 | 428.9 | 136.9 | 274.2 | 76.6  | 160.9 | 62.1  | 130.6 | 31.6  | 66.7  | 25.2  | 53.2  | 52.6  | 39.2  |
| Ns. d1 | 208.9 | 333.9 | 143.4 | 207.9 | 81.2  | 117.7 | 66.4  | 94.3  | 34.7  | 46.1  | 27.9  | 36.3  | 38.8  | 27.9  |
| Ns. d2 | 243.4 | 354.8 | 167.3 | 218.7 | 95.0  | 122.8 | 77.8  | 98.0  | 40.8  | 47.3  | 32.9  | 37.1  | 46.2  | 33.7  |
| Ns. d3 | 216.6 | 347.7 | 147.4 | 213.4 | 83.3  | 119.4 | 68.0  | 95.1  | 35.3  | 45.7  | 28.3  | 35.7  | 46.0  | 33.5  |
| Ns. d4 | 224.6 | 363.2 | 153.4 | 224.6 | 87.4  | 126.7 | 71.5  | 101.2 | 37.4  | 49.2  | 30.1  | 38.6  | 50.4  | 37.1  |
| Ns. d5 | 222.4 | 371.0 | 150.9 | 228.3 | 84.7  | 128.1 | 69.0  | 102.2 | 35.6  | 49.3  | 28.5  | 38.6  | 42.9  | 31.5  |
| Ns. d6 | 244.4 | 415.5 | 166.2 | 255.5 | 93.6  | 143.4 | 76.3  | 114.3 | 39.5  | 55.1  | 31.7  | 43.2  | 50.5  | 37.1  |
| Ns. d7 | 264.2 | 457.9 | 179.7 | 279.8 | 101.0 | 155.9 | 82.3  | 123.9 | 42.6  | 59.2  | 34.2  | 46.2  | 43.4  | 31.9  |
| Ns. d8 | 303.0 | 489.5 | 206.8 | 297.8 | 117.4 | 165.4 | 95.9  | 131.3 | 50.1  | 62.4  | 40.3  | 48.7  | 55.6  | 40.9  |

**Table S2: The scattering (S) and absorption (A for the CLAP and AE for the aethalometer) coefficient (Mm$^{-1}$) of tar balls at different wavelengths served as input parameters for the calculation of the index of refraction.**

\* indicate that the data were intra- or extrapolated

Bl. – Black locust

Ns. – Norway spruce

---

## Author Response (AR2)

**Response to the Associate Editor**

**Associate Editor Decision: Reconsider after major revisions** (11 Apr 2017) by Alexander
Kokhanovsky

**Minor comments:**

1. *Fig.2 must be improved ( see your caption for axis OX; also for clarity I would suggest that you have just 2 lines in this figure - say, solid and broken)*

The figure has been corrected according the suggestion and it has been replaced.

2. *Fig.3. Please, check the caption on the axis OX in both panels.*

The x axis has been modified on both panels, the corrected figure is placed to the manuscript.

3. *Fig.4a. I would suggest to have the same caption on axis OX as in Fig.3. Also I would propose to change the caption for axis OY ( ratio of absorption coefficients ( say, r)). I do not understand why r for the red line is not equal to 1.0 at 450nm. Please, explain in the caption.*

This should be a misunderstanding. There is no Figure 4a in our paper whatsoever.  The referred Figure 4a was in our previous ACPD submission that was sent to the editor in a separate e-mail on 5$^{th}$ April 2017 upon his request.  This particular figure in the ACP manuscript referred to AERONET data interpretation that is not part of our AMT manuscript.

**Major comments:**

*You state that the conventional belief is that tar balls are not strong absorbers in the near IR. I would suggest that you extend your Introduction by analyzing some papers, which report smaller absorption by brown carbon in near - IR and also pointing to the possible reasons for the differences with your results.*
*In particular, I would suggest that you review the results of the following papers in Introduction:*
*1) ) Laskin et al. (2015): Chemistry of Atmospheric Brown Carbon, Chem. Rev. 115, 4335−4382*

[revised manuscript text omitted]